# Enhanced Stability of Detergent-Free Human Native STEAP1 Protein from Neoplastic Prostate Cancer Cells upon an Innovative Isolation Procedure

**DOI:** 10.3390/ijms221810012

**Published:** 2021-09-16

**Authors:** Jorge Barroca-Ferreira, Pedro Cruz-Vicente, Marino F. A. Santos, Sandra M. Rocha, Teresa Santos-Silva, Cláudio J. Maia, Luís A. Passarinha

**Affiliations:** 1CICS-UBI–Health Sciences Research Centre, University of Beira Interior, 6201-506 Covilhã, Portugal; jorgedanielferreira@gmail.com (J.B.-F.); pedromvcruz@hotmail.com (P.C.-V.); sandracmrocha@gmail.com (S.M.R.); cmaia@fcsaude.ubi.pt (C.J.M.); 2Associate Laboratory i4HB-Institute for Health and Bioeconomy, NOVA School of Science and Technology, Universidade NOVA de Lisboa, 2819-516 Caparica, Portugal; mf.santos@campus.fct.unl.pt (M.F.A.S.); tsss@fct.unl.pt (T.S.-S.); 3UCIBIO–Applied Molecular Biosciences Unit, Department of Chemistry, NOVA School of Science and Technology, Universidade NOVA de Lisboa, 2819-516 Caparica, Portugal; 4Laboratório de Fármaco-Toxicologia-UBIMedical, University of Beira Interior, 6201-284 Covilhã, Portugal

**Keywords:** circular dichroism, co-immunoprecipitation, prostate cancer, protein purification, STEAP1, thermal stability

## Abstract

Background: The STEAP1 is a cell-surface antigen over-expressed in prostate cancer, which contributes to tumor progression and aggressiveness. However, the molecular mechanisms underlying STEAP1 and its structural determinants remain elusive. Methods: The fraction capacity of Butyl- and Octyl-Sepharose matrices on LNCaP lysates was evaluated by manipulating the ionic strength of binding and elution phases, followed by a Co-Immunoprecipitation (Co-IP) polishing. Several potential stabilizing additives were assessed, and the melting temperature (*T*m) values ranked the best/worst compounds. The secondary structure of STEAP1 was identified by circular dichroism. Results: The STEAP1 was not fully captured with 1.375 M (Butyl), in contrast with interfering heterologous proteins, which were strongly retained and mostly eluted with water. This single step demonstrated higher selectivity of Butyl-Sepharose for host impurities removal from injected crude samples. Co-IP allowed recovering a purified fraction of STEAP1 and contributed to unveil potential physiologically interacting counterparts with the target. A *T*m of ~55 °C was determined, confirming STEAP1 stability in the purification buffer. A predominant α-helical structure was identified, ensuring the protein’s structural stability. Conclusions: A method for successfully isolating human STEAP1 from LNCaP cells was provided, avoiding the use of detergents to achieve stability, even outside a membrane-mimicking environment.

## 1. Introduction

The Six-Transmembrane Epithelial Antigen of the Prostate 1 (STEAP1) is an integral membrane protein composed of six-transmembrane helices located in both tight- and gap-junctions, cytoplasm, and endosomal membranes, connected by intra- and extra-cellular loops [1]. STEAP1 is particularly over-expressed in prostate cancer (PCa), in contrast with non-tumoral tissues and vital organs, unveiling its specificity for cancer microenvironments [2]. Based on amino-acid sequence, transmembrane topology, and cellular localization, it was hypothesized that STEAP1 may play an important role as a transporter protein being involved in cell communication [3] and in the stimulation of cell growth by increasing the levels of reactive oxygen species [4]. The recent cryo-EM structure-function analysis of STEAP1 cloned in Human Embryonic Kidney (HEK) cells and bound to an antibody-fragment (6Y9B, 2.97 Å resolution) revealed a trimeric arrangement supporting its functional role in heterodimeric assembles with other STEAP1 paralogs to recruit and orient intracellular electron-donating substrates to enable transmembrane-electron transport and the reduction of extracellular metal-ion complexes [5,6].

The over-expression of STEAP1 enhances cancer cell proliferation and contributes to tumor development and aggressiveness [7,8]. Regarding the regulation of STEAP1 in PCa, it was demonstrated that 5α-dihydrotestosterone (DHT) down-regulates STEAP1 expression in LNCaP cells by androgen-independent receptor, suggesting that this down-regulation occurs in response to proliferation effects of DHT [9]. Moreover, it was shown that the knock-down of STEAP1 abrogated the proliferation and anti-apoptotic effects of DHT, indicating that blocking the STEAP1 protein can be advantageous in the treatment of hormone-dependent PCa [10]. Indeed, several studies with monoclonal antibodies attached to radioisotopes have demonstrated promising results in targeting and monitoring STEAP1 expression and in controlling PCa progression [11,12]. Likewise, several in vitro and in vivo studies revealed that STEAP1-derived peptides are immunogenic and hence suitable for recognition by cytotoxic T lymphocytes [13,14], suggesting their potential use in the development of anti-cancer vaccines. Altogether, these features highlight the usefulness of STEAP1 as a promising tool, either as a biomarker or as a target for anti-cancer therapies [15,16].

The STEAP1 is the most relevant member of the STEAP family from a clinical perspective [17]. Currently, only two crystal structures of the membrane-proximal oxidoreductase domain human STEAP3 (2VNS, 2 Å resolution and 2VQ3, 2 Å resolution) [18] and two structures of human STEAP4 domains (6HD1, 3.8 Å resolution and 6HCY, 3.1 Å resolution) [19] are deposited in the Protein Data Bank (PDB, 6th September 2021). To decipher the molecular interactions between STEAP1 and specific molecules, namely for designing selective antagonist drugs capable of blocking its oncogenic role, a full characterization of the STEAP1 protein is required. One of the major drawbacks associated with structure-based design studies relies on attaining high amounts of the biological target with substantial purification yields. To overtake these issues, the expression system must be carefully chosen, and a proper isolation strategy should be designed and fully optimized. Despite the challenges in handling membrane proteins (MPs), over the last few years, our research group has successfully focused on the in-depth optimization of up- and down-stream processing conditions [20,21]. So far, there are no native STEAP1 high-resolution structures available mostly due to the difficulty in obtaining high amounts of structured protein from LNCaP cells.

Furthermore, the purification of MPs involves several sequential techniques exploring not only the intrinsic properties of MPs, but also considering the compatibility with a solubilizing detergent. Recently, we proposed the application of a glycerol gradient fed-batch profile associated with a methanol constant feed, supplemented with 6% (*v/v*) DMSO and 1 M Proline as an ideal fermentation strategy to improve the biosynthesis and stabilization of biologically active recombinant human STEAP1 in mini-bioreactor *Komagataella pastoris* X-33 Mut^+^ cultures [22]. However, further isolation and purification reports of this STEAP1 counterpart are still not available in the literature. To the best of our knowledge, there are only two studies focused on recombinant STEAP1 (expressed in HEK and Baculovirus-Insect cells) isolation using Immobilized Metal Affinity Chromatography (IMAC) followed by Size Exclusion Chromatography (SEC) [5,6]. These works, although very informative, lack experimental data on STEAP1 derived from native source namely its putative structure and chemical modifications. Altogether, these facts prompt new research paths exploring the production, extraction, and purification of STEAP1 from its natural cancer microenvironment to subsequently characterize its thermal stability and structural rearrangement.

In this work, we successfully extracted and purified the native full-length human STEAP1 protein from LNCaP prostate cancer cells by exploring two traditional hydrophobic matrices—Butyl- and Octyl-Sepharose—enhancing their chromatographic behavior and performance. Moreover, an innovative polishing step of the pre-purified sample from Hydrophobic Interaction Chromatography (HIC) using Co-Immunoprecipitation (Co-IP), was included allowing to unveil potential STEAP1-interacting moieties. The strategy here adopted represents a novelty in separation and purification of MPs and could be applied to other members of this large class of proteins. Thereafter, the obtained native STEAP1 sample was used to gain insights regarding its biophysical and structural properties by Thermal Stability Assay (TSA) and UV Circular Dichroism (CD) Spectroscopy, respectively.

## 2. Results and Discussion

MPs play crucial roles in a wide variety of cellular functions and represent >50% of currently marketed therapeutic targets, highlighting their importance in structural biology. Despite their relevance, MPs represent less than 2% of the deposited structures in the PDB (https://www.rcsb.org/, (accessed on 6 September 2021)). Therefore, the structural and functional characterization of these proteins could provide details on their mechanism of action which is relevant for the rational design of novel drugs. The disparity between the available information for MPs and soluble counterparts is explained by their physiochemical properties, particularly their increased hydrophobicity and low natural abundance, hampering the production steps from expression to purification and, ultimately, affecting the respective characterization [23,24].

Considering the clinical relevance of STEAP1 and its potential application as a promising therapeutic agent against PCa, it is crucial to explore several expression systems, extraction strategies, and purification approaches to ultimately obtain a high-resolution 3D structure of the protein. Despite several attempts with the recombinant isoform (Table 1), experimental data about STEAP1 from its native cancer environment are still inexistent.

Our first goal was to select an appropriate procedure for total protein extraction from the LNCaP prostate cancer cell line, which naturally expresses high levels of STEAP1 in its native conformation, based on a liquid chromatography system. The initial purification of MPs is commonly conducted by IMAC using fusion tags, usually polyhistidine tags, often coupled to SEC to separate proteins based on their hydrodynamic volume, which may directly correspond to molecular weight [23,24]. In turn, HIC is a powerful and widespread separation technique in lab-scale purification and has been extensively explored as an alternative technique for the purification of biomolecules, and a major step in downstream processing, often yielding highly pure MPs for biomedical applications [25]. The native structure of STEAP1 presents transmembrane helices in the internal core and an anchoring region—a 69 residue N-terminal intracellular tail [1]. These hydrophobic features enable a highly specific interaction between STEAP1 and HIC matrices, even when embedded in a complex biological mixture.

Therefore, we fully refined a HIC workflow, thoroughly screening typical chromatographic parameters: buffer composition, ionic strength, and resin properties (please see Appendix A [25,26,27,28,29,30,31,32,33,34]). Briefly, a wide range of (NH_4_)_2_SO_4_ concentrations in the binding buffer were evaluated as this salt affects the exposure of the hydrophobic moieties of STEAP1, which will further interact with the chromatographic matrices here tested, Butyl- and Octyl-Sepharose, to form a protein-ligand complex [25]. By manipulating the ionic strength, we intended to obtain a single protein faction minimum, co-eluting interfering compounds. Indeed, 1.375 M (NH_4_)_2_SO_4_ was revealed to be the optimal salt concentration for considerable amounts of pure STEAP1 in the flowthrough, using Butyl-Sepharose resin (Figure 1, Peak I), with an estimated concentration of 50 µM upon pre-purification trials from the LNCaP cells crude extract.

Moreover, we concluded that Butyl-Sepharose resin presents an increased selectivity for impurities removal of the initial sample from LNCaP extracts without affecting the yield of STEAP1 isolation, resembling a negative chromatography-like approach. In this operation mode, contaminants are initially adsorbed onto the resin and the target protein is recovered in the flowthrough pools. This result could also be explained considering the tridimensional reorientation of the protein which influences the interaction with the chromatographic resin: if the hydrophobic intracellular tail of STEAP1 is hidden, a possible interaction between the protein and the column is circumvented and, consequently, the contaminants will strongly interact with the matrix instead. Another explanation may be due to a structural rearrangement of STEAP1 to a most stable conformation in order to compensate the absence of detergent in the binding and elution buffers, since these tensoactive agents are reported as crucial for MPs reconstitution [35]. However, the use of surfactants in this early phase of the STEAP1 isolation was discarded to not jeopardize further characterization techniques which, often, require their posterior removal by dialysis or similar methodologies. So far, negative chromatography was already used for the purification of antibodies [36,37], recombinant proteins [38,39] and virus-like particles [40,41] exhibiting better purity and recovery performances than other commonly reported techniques and surpassing the binding capacity limitation of typical chromatographic matrices [42].

Despite the high quality of the pre-purified extract, we aim to obtain a pure and stable STEAP1 sample. An additional SEC step was firstly considered, but the amount of the recovered sample might be drastically reduced, threatening further characterization trials. Alternatively, Co-IP was an attractive highly specific and selective option for the isolation of STEAP1 from the pre-purified extract as well as a polishing step to remove the remaining residual contaminants. This approach is effective in the isolation of a specific antigen from complex samples through a non-covalently immobilized antibody onto cross-linked agarose beads [43,44]. Likewise, Co-IP will be also important in the identification of biomolecules that may directly or indirectly interact with STEAP1. Co-IP has become a popular method and a powerful tool for: (i) disclosing protein–protein interactions which regulate several intra- and intercellular biological processes [45,46]; (ii) identifying the formation of protein-ligand complexes [47,48]; (iii) evaluating the differential expression of a protein; and (iv) determining the molecular weight and posttranslational modifications of proteins [49,50]. Indeed, SDS-PAGE electrophoresis showed a high purity of the co-immunoprecipitated extract, through a considerable reduction of interfering proteins, when compared to the pre-purified STEAP1 from Butyl-Sepharose. Furthermore, the WB analysis revealed the presence of STEAP1 in the immunoprecipitated sample with the predicted molecular weight (~35 kDa) as well as increased molecular weight bands (~75–100 kDa), probably corresponding to STEAP1 aggregates or unspecific interactions (major detection at ~63 kDa, protein not identified) (Figure 2).

The stability of STEAP1 sample obtained by Co-IP was properly evaluated by TSA, a high-throughput screening method quite effective in assessing protein stability under different conditions, following the protein thermal denaturation process, and determining the respective melting temperature (*T*m) [51,52]. As the temperature increases, the protein is partially or fully unfolded exposing the hydrophobic core that interacts with a fluoroprobe, leading to an increased fluorescence signal while the structural integrity is continuously monitored [53]. A first study was performed to evaluate the thermostability of STEAP1 in the initial purification buffer (10 mM Tris-HCl, pH 7.8). The denaturation curve was analyzed and the first derivative was calculated revealing a two-state unfolding model with two distinct sharp peaks with *T*m ~41 °C and *T*m ~55 °C; both peaks likely correspond to STEAP1 denaturation, and the first transition might represent the denaturation of a particular domain. The higher *T*m indicates that the protein assumes a stable conformational state, suggesting the unfolding of highly energetically coupled multi-domains or different populations of proteins (Figure 3) [54,55].

These data could be justified considering the SDS-PAGE profile of the co-immunoprecipitated sample by which STEAP1 might interact with other counterparts to form protein-protein complexes (Figure 2). Curiously, this result resembles a recent STEAP1 thermostability study reporting a *T*m of ~55.5 °C [5]. TSA is also a very powerful tool to evaluate the best protein buffer formulation as well as to find promising additives, ligands, and other small molecules that improve protein homogeneity, solubility, stability, purification, and storage, which might facilitate STEAP1 crystallization and the respective structural characterization [56]. Using a 96-conditions additive screening (RUBIC Additive Screen Kit), we aimed to identify putative additives able to increase the STEAP1 thermal stability. The *T*m values were determined and the ones corresponding to a single transition state were further compared with the control value (*T*m of 58.74 °C). Interestingly, the thermal denaturation profile of both sodium phosphate buffer mono- (*T*m of 50.18 °C) and dibasic (*T*m of 46.60 °C) exhibit a drastic Tm reduction, resulting in a non-analytical melting curve (Figure 4A). These results undoubtedly justify the disposal of phosphate buffer dual-salt system and their impacting effects in STEAP1 structural integrity throughout the downstream processing. Furthermore, we also evaluated the effect of several non-detergents and detergents to confirm if these membrane lipid mimicking agents are truly needed to solubilize STEAP1. The results revealed that none of the tested reagents—Non-Detergent Sulfobetaines (NDSB)-195 (*T*m of 58.21 °C), NDSB-201 (*T*m of 58.25 °C), CHAPSO (*T*m of 57.57 °C), or Octyl-β-Glucoside (OG) (*T*m of 58.46 °C)—conferred a significant *T*m increase when compared to the control condition (Figure 4B) suggesting that the presence of detergents has no effect on the protein stability and can be circumvented. This is of utmost importance for drug-design campaigns, since surfactants can block protein-ligand binding sites, impairing the discovery of novel therapeutic drugs. Likewise, it also suggests that the proposed recovery of an active and fully solubilized fraction of STEAP1, maintaining its epitope throughout its isolation, is possible with no need of additional use of detergents.

Moreover, the additives Gly-Gly-Gly (*T*m of 59.72 °C), PEG3350 (*T*m of 59.64 °C), DNA Library (*T*m of 59.53 °C), Biotin (*T*m of 59.31 °C), and TCEP (*T*m of 59.29 °C) exhibit higher *T*m values than the control (Figure 4C). The observed slight positive melting temperature shift (Δ*T*m) might contribute to increase the protein stability reducing the respective conformational flexibility favoring further structural studies [57,58]. Hence, the referred additives could be considered promising candidates to be included in (i) the early stages of STEAP1 production and expression to promote a proper folding and to prevent aggregation, and (ii) the final protein buffer as putative crystallization additives. Several other additives noticeably decreased the STEAP1 thermal stability, in particular Fos Choline 12 (*T*m of 36.09 °C), K^+^ Sulfate (*T*m of 44.08 °C), Na^+^ Phosphate (dibasic) (*T*m of 46.60 °C), Mg^2+^ Sulfate (*T*m of 48.08 °C) and Na^+^ Phosphate (monobasic) (*T*m of 50.18 °C). These negative Δ*T*m values potentially indicate important structural changes towards a more disordered conformation or even protein misfolding [57,58]. A ranking of the best and the worst additives and their respective Tm is summed up in Figure 5.

Furthermore, CD spectroscopy allowed ascertaining the secondary structure of human native STEAP1, and to determine whether the structural stability and the correct folding of the protein of interest was upheld throughout the isolation procedure. The availability of methods for structural characterizing MPs is paramount. CD spectroscopy is a powerful biophysical tool that provides useful insights on the protein secondary structure and potential conformational changes [59]. The results showed that STEAP1 spectrum features an intense negative band at 209 nm and a positive band at 196 nm (Figure 6).

This typical CD signature revealed for the first time that the native human STEAP1 predominantly adopts an α-helical structure [60,61]. The slight wavelength shift of the maximum value when compared to typical absorbances for α-helix structures (193 nm), was attributed to the large hydrophobic nature of STEAP1 and the environment wherein the protein of interest is sequestered [60,61]. This finding is in full concordance with the predicted α-helical transmembrane arrangement cryo-EM structure of trimeric human STEAP1 bound to three Fab fragments, already deposited in the PDB [5].

## 3. Materials and Methods

### 3.1. Chemicals

Ultrapure reagent-grade water was obtained from a Milli-Q system (Millipore/Waters, Etten-Leur, The Netherlands). Ammonium sulfate, ethylenediamine tetra-acetic acid (EDTA) and sodium dodecyl sulfate (SDS) were obtained from PanReac Applichem (Darmstadt, Germany). Hydrochloric acid and Tween-20 were purchased from ThermoFisher Scientific (Waltham, MA, USA). Sodium chloride was obtained from Honeywell (Charlotte, NC, USA) Sodium deoxycholate was acquired from Sigma-Aldrich Co. (St. Louis, MO, USA). Tris-base was obtained from Fisher Scientific (Epson, UK). Nonidet P-40 (NP-40) was purchased from Fluka (Monte Carlo, Monaco). Bis-Acrylamide/Acrylamide 40% was bought from GRiSP Research Solutions (Oporto, Portugal). The NZYColour Protein Marker II was acquired from NZYTech (Lisbon, Portugal). All chemicals were of analytical-grade commercially available and used without further purification.

### 3.2. LNCaP Cell Culture

The LNCaP prostate cancer cell line was purchased from the European Collection of Cell Cultures (ECACC, UK) and maintained in RPMI 1640 medium (Sigma-Aldrich Co. St. Louis, MO, USA), supplemented with 10% fetal bovine serum (FBS) (Biochrom AG, Berlin, Germany) and 1% penicillin/streptomycin (ThermoFisher Scientific, Waltham, MA, USA), in a humidified chamber at 37 °C and a 5% CO_2_ atmosphere. LNCaP cells have grown in 75 cm^2^ t-flasks (*n* = 6) until 80–90% confluence for further harvest.

### 3.3. Cell Lysis and Total Protein Quantification

LNCaP cells were lysed on an appropriate volume of RIPA buffer (150 mM NaCl, 1% NP-40, 0.5% sodium deoxycholate, 0.1% SDS, 50 mM Tris, 1 mM EDTA) supplemented with 1% protease inhibitors cocktail (Hoffmann-La Roche, Basel, Switzerland) and 10% Phenylmethylsulfonyl Fluoride (PMSF) (PanReac Applichem, Darmstadt, Germany). The total protein extract was obtained after centrifugation of the cell lysate for 20 min at 16,000 rpm and 4 °C. Quantification of the total amount of protein was measured using Pierce™ BCA Protein Assay Kit (ThermoFisher Scientific, Waltham, MA, USA).

### 3.4. Hydrophobic Interaction Chromatography

The isolation trials were performed in an ÄKTA Avant system with UNICORN 6 software (GE Healthcare, Wauwatosa, WI, USA) at room temperature. All buffers were filtered through a 0.22 μm pore size membrane and ultrasonically degassed. Butyl-Sepharose 4FF and Octyl-Sepharose 6 FF (GE Healthcare, Wauwatosa, WI, USA) were used as HIC stationary phases. The hydrophobic matrices were packed according to the company guidelines (10 mL gel volume packed into an XK-16 glass column, GE Healthcare, Wauwatosa, WI, USA). The columns were equilibrated with the different tested concentrations of ammonium sulfate in Tris-HCl 10 mM, pH 7.8. LNCaP total protein extracts (500 µL with a protein concentration of ~12 mg/mL) were loaded onto the columns. Isocratic elution at 1.0 mL/min was performed by decreasing stepwise the ammonium sulfate concentration up to 0 M. The pH, pressure, conductivity, and 280 nm absorbance were continuously monitored throughout the entire chromatographic run. The fractions of interest were collected, desalted, concentrated, and stored at 4 °C to further purity and immunoreactivity analysis. The protein content of HIC fractions was measured by Pierce™ BCA Protein Assay Kit (ThermoFisher Scientific, Waltham, MA, USA).

### 3.5. Co-Immunoprecipitation

Co-IP protocol was conducted according to the instructions of the manufacturer for Protein A/G PLUS-Agarose Immunoprecipitation Reagent (sc-2003, Santa Cruz Biotechnology, Dallas, TX, USA) with some in-lab optimizations. Briefly, the pre-purified sample of STEAP1 obtained from HIC was incubated overnight at 4 °C with anti-STEAP1 mouse monoclonal antibody (B-4, Santa Cruz Biotechnology, Dallas, TX, USA) followed by an additional overnight incubation with agarose beads with constant stirring at 4 °C. The immunoprecipitates were collected by centrifugation at 16,000× *g* for 5 min at 4 °C. The supernatant was discarded, and the pellet was washed and resuspended in the electrophoresis sample. The protein was then recovered from agarose beads due to the combined action of reducing agent β-mercaptoethanol and 100 °C temperature.

### 3.6. SDS-PAGE and Dot-/Western-Blot

Reducing SDS-polyacrylamide gel electrophoresis (SDS-PAGE) was carried out according to the method of Laemmli [62]. Samples were boiled for 5 min at 100 °C and resolved in two 10% SDS-PAGE gels at 120 V for approximately 2 h. Then, one gel was stained by Coomassie brilliant blue and the second gel was transferred into a polyvinylidene difluoride (PVDF) membrane (GE Healthcare, Wauwatosa, WI, USA) at 750 mA for 90 min and 4 °C. Approximately 100 µL of samples were injected onto PVDF membranes for dot-blot analysis. Membranes were blocked for 1 h in a 5% (*w/v*) non-fat milk solution in TBS-T and incubated overnight with anti-STEAP1 mouse monoclonal antibody 1:100 (B-4, sc-271872, Santa Cruz Biotechnology, Dallas, TX, USA) at 4 °C with constant stirring. After, membranes were incubated with goat anti-mouse IgG-HRP 1:20,000 (sc-2005, Santa Cruz Biotechnology, Dallas, TX, USA) for 2 h at room temperature with constant stirring. Finally, STEAP1 immunoreactivity was visualized using ChemiDoc™ MP Imaging System (Bio-Rad, Hercules, CA, USA) after a brief incubation with Chemiluminescent HRP substrate (Merck, Germany).

### 3.7. Thermal Shift Assay

The TSA was performed to assess protein stability upon incubation in purification buffer (10 mM Tris-HCl, pH 7.8). The assays were completed in MicroAmp^®^Fast 96-well reaction plates (ThermoFisher Scientific, Waltham, MA, USA), using a total sample volume of 20 μL, containing 17 μL of STEAP1 at 30 μM and 3 μL of Protein Thermal Shift™ Dye Kit (ThermoFisher Scientific, Waltham, MA, USA). A second TSA experiment was conducted to disclose the effects of several buffers and additives in the stabilization of STEAP1 through RUBIC Additive Screen (Molecular Dimensions, Maumee, OH, USA), according to manufacturers’ instructions. The assay was performed in MicroAmp^®^Fast 96-well reaction plates (ThermoFisher Scientific, Waltham, MA, USA), using a total sample volume of 20 μL, containing 10 μL of each RUBIC ligand, 5 μL of 4× protein purification buffer (10 mM Tris-HCl, pH 7.8), 2 μL of STEAP1 at 30 μM and 3 μL of Protein Thermal Shift™ Dye Kit (ThermoFisher Scientific, Waltham, MA, USA). The TSA was performed in 2 min cycles of 1% increments between 25 °C and 95 °C in a StepOnePlusTM Real-Time PCR System (ThermoFisher Scientific, Waltham, MA, USA). Data processing and analysis were performed with Protein Thermal Shift^TM^ Software (ThermoFisher Scientific, Waltham, MA, USA). The melting temperature values were determined using the first derivative values of the raw fluorescence data.

### 3.8. Circular Dichroism Spectroscopy

CD Spectroscopy was applied to assess the secondary structure of purified STEAP1. CD spectra were acquired in a Jasco J-815 spectrometer (Jasco, Easton, MD, USA), using a Peltier temperature controller (model CDF-426S/15). Spectra were recorded using a 1 mm path-length quartz cuvette with a STEAP1 concentration of 30 μM in purification buffer (10 mM Tris-HCl, pH 7.8). The data represent the average of 3 scans from 250 to 190 nm with a step size of 0.1 nm and a response time of 1 s. The CD spectrum of the buffer (10 mM Tris-HCl, pH 7.8) was collected at room temperature and was used as a blank. The CD spectra for STEAP1 were corrected for background and baseline with the buffer blank.

## 4. Conclusions

In this work, we established, for the first time, a reproducible chromatographic procedure coupled to Co-IP for the isolation of full-length human native STEAP1 obtained from LNCaP extracts. A protein sample was pre-purified using a traditional Butyl-Sepharose hydrophobic matrix with 1.375 M (NH_4_)_2_SO_4_ as buffer, allowing recovery of the STEAP1 protein in a single step throughout binding phase with residual interfering proteins and avoiding the use of detergents or other membrane mimicking agents. The Co-IP method was further used as an effective and novel polishing strategy for achieving a sample of STEAP1 with a significant purity degree. According to TSA, the recovered purified native human STEAP1 dismisses the addition of a specific buffer or additive to achieve stability, even outside its native membrane environment. Moreover, the CD trial confirmed that STEAP1’s predominantly α-helical secondary structure was preserved during the solubilization and purification events. The flowsheet proposed here will be the basis to obtain the crystal structure of STEAP1 in a stable, functional, and native state, which remains to be determined.

## Figures and Tables

**Figure 1 ijms-22-10012-f001:**
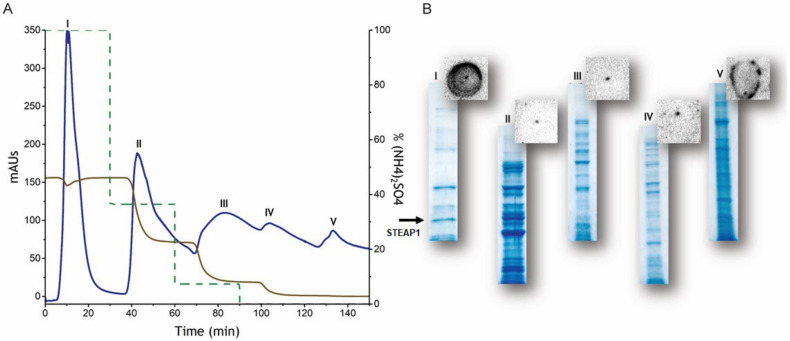
(**A**) Chromatographic profile of STEAP1 purification on Butyl-Sepharose 4 FF resin with optimized conditions. Blue line represents absorbance at 280 nm. Adsorption performed at 1.375 M (NH_4_)_2_SO_4_ in 10 mM Tris-HCl buffer at pH 7.8 (1.0 mL min^−1^). Desorption was performed at 500 mM and 100 mM (NH_4_)_2_SO_4_, 10 mM Tris-HCl and a final step with H_2_O (1.0 mL min^−1^), in stepwise gradient. (**B**) SDS-PAGE and dot-blot analysis depicted for each peak. The gel was stained with Coomasie brilliant blue and the membranes were incubated with anti-STEAP1 mouse primary antibody overnight at 4 °C, followed by goat anti-mouse secondary antibody incubation at room temperature. STEAP1 is marked with a black arrow.

**Figure 2 ijms-22-10012-f002:**
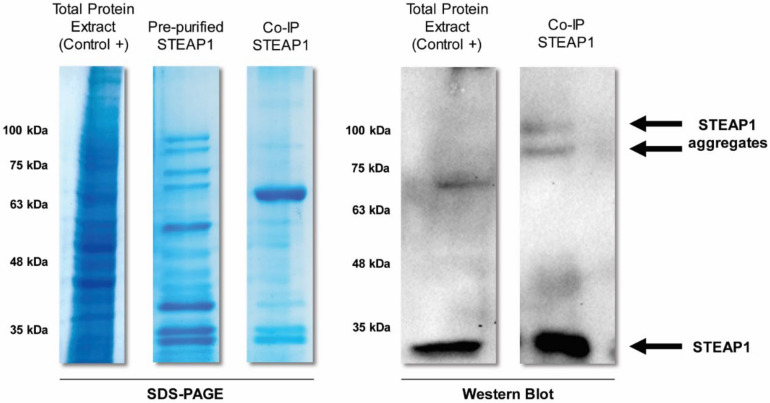
SDS-PAGE and WB analysis of STEAP1 and its aggregates obtained in each step of the overall bioprocessing: the initial total protein extract from LNCaP cells, the pre-purified sample from Butyl-Sepharose 4FF and the final co-immunoprecipitated extract. The gel was stained with Coomasie brilliant blue and the membranes were incubated with anti-STEAP1 mouse primary antibody overnight at 4 °C followed by goat anti-mouse secondary antibody incubation at room temperature.

**Figure 3 ijms-22-10012-f003:**
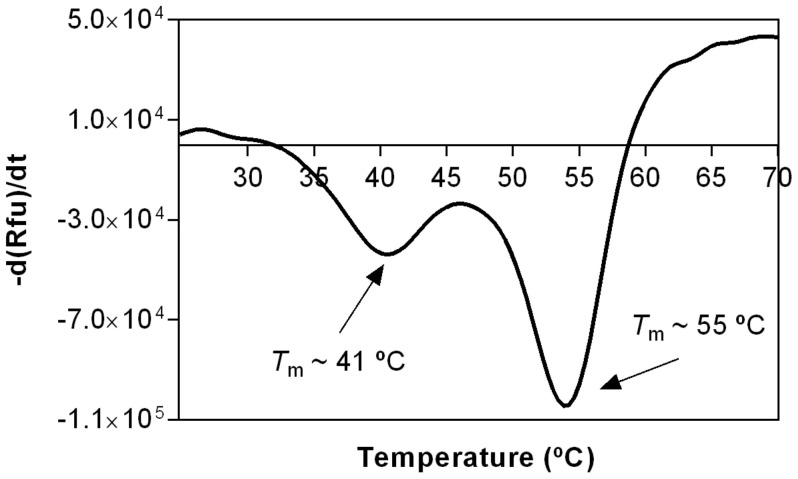
TSA melting curve of STEAP1 obtained from thermal stability fluorescence data of first derivative (d(Rfu)/dt) curves in 10 mM Tris-HCl, pH 7.8 buffer exhibiting a *T*m of 41 and 55 °C.

**Figure 4 ijms-22-10012-f004:**
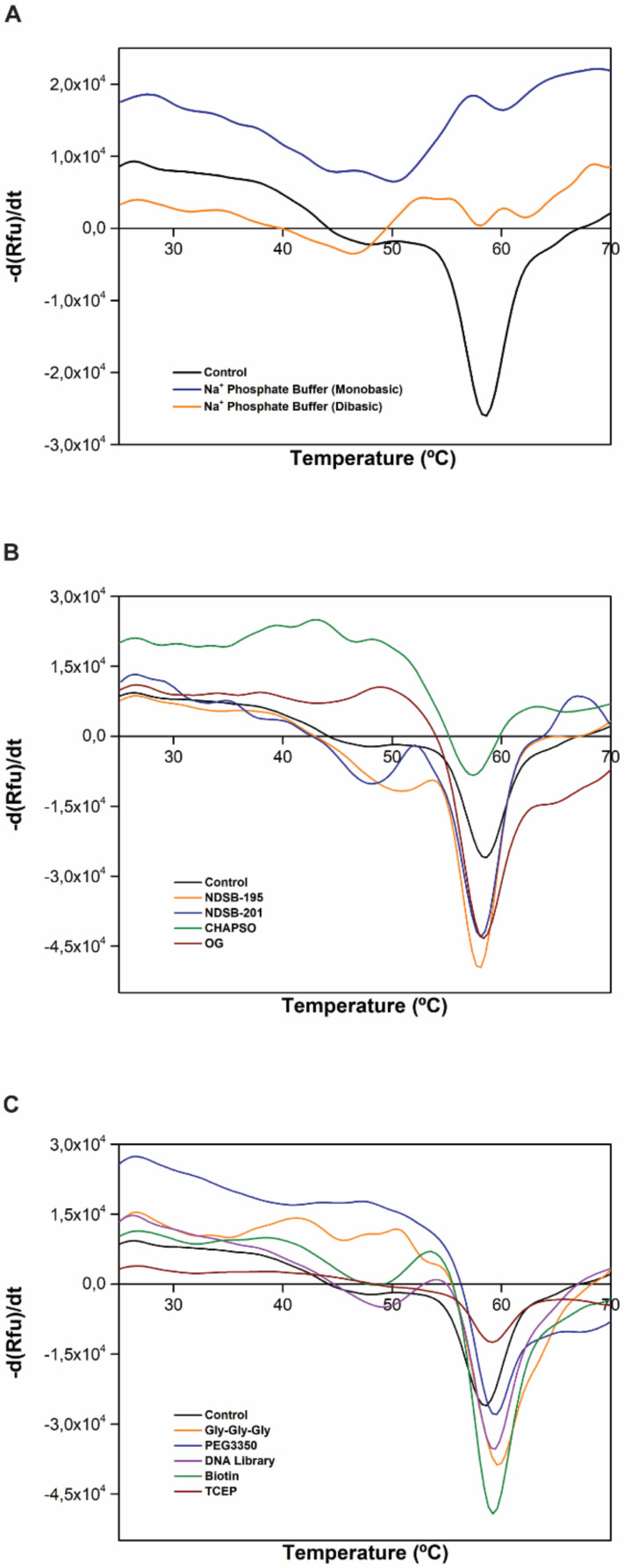
TSA melting curves of STEAP1 obtained from thermal stability fluorescence data of first derivative (d(Rfu)/dt) with RUBIC Additive Screen Kit for the control experiment, exhibiting a *T*m of 58.74 °C and (**A**) in sodium phosphate buffer mono- and dibasic; (**B**) in non-detergents (NDSB-195 and NDSB-201, with a *T*m of 58.21 °C and 58.25 °C, respectively) and detergents (CHAPSO and OG, with a *T*m of 57.57 and 58.46 °C, respectively); (**C**) in best thermal stabilizer additives (Gly-Gly-Gly, PEG3350, DNA Library, Biotin, and TCEP, with a *T*m of 59.72, 59.64, 59.53, 59.31, and 59.29 °C, respectively).

**Figure 5 ijms-22-10012-f005:**
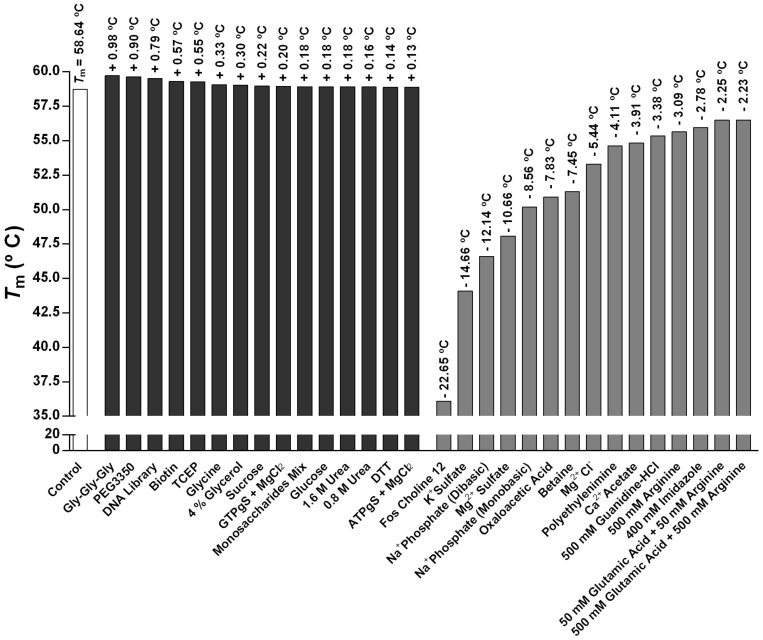
Graphical representation of the Tm values of STEAP1 in the presence of best (black) and least impressive (dark grey) additives from RUBIC Additive Screen Kit. The control experiment (white) was prepared using water.

**Figure 6 ijms-22-10012-f006:**
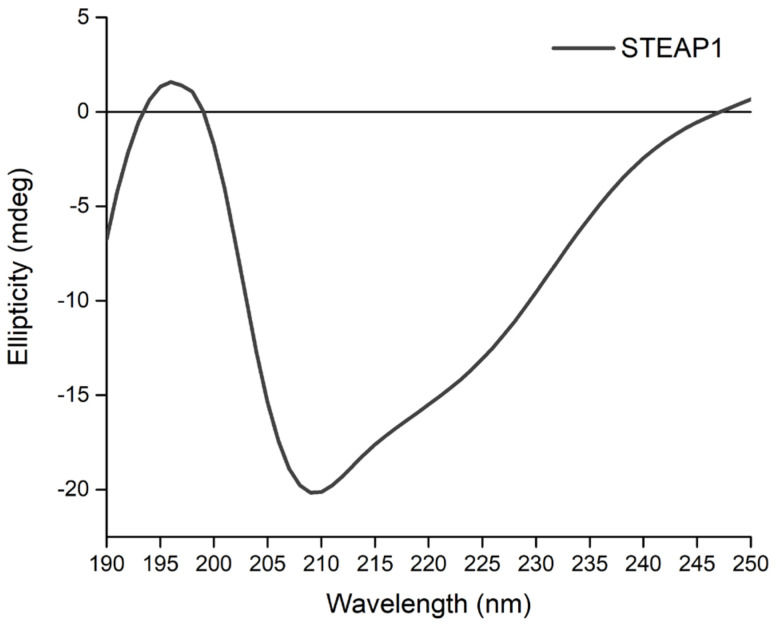
Representative CD spectrum of secondary structure of STEAP1 in 10 mM Tris-HCl, pH 7.8 buffer showing a minimum and a maximum peak at 209 and 196 nm, respectively.

**Table 1 ijms-22-10012-t001:** Integrative overview of overall existent strategies from the biosynthesis to the isolation of STEAP1 counterparts towards their structural resolution and characterization.

Protein	Expression System	Extraction	Isolation	Chromatographic Buffers	Structural Resolution	Ref.
Native Human STEAP1	Neoplastic Prostate Cancer Cells (LNCaP)	RIPA Buffer (50 mM Tris Base, 150 mM NaCl, 1 mM EDTA, 0.5% Sodium Deoxycholate, 0.1% SDS, 1% NP-40, pH 7.8)	Hydrophobic Interaction Chromatography (Butyl-Sepharose) coupled to Co-Immunoprecipitation	1.375 M (NH_4_)_2_SO_4_, pH 7.8 (Binding) 10 mM Tris, pH 7.8 (Elution)	n.a.	This Work
Recombinant Human STEAP1	Human Embryonic Kidney Cells (HEK)	Ordinary Lysis Buffer (50 mM Tris, 250 mM NaCl, 0.7% digitonin, 0.3% n-Dodecyl-β-D-Maltoside, 0.06% Cholesteryl hemi-succinate, pH 7.8)	Affinity Chromatography (Streptactin) (*A*) Size Exclusion Chromatography (Superdex 200 10/300) (*B*)	50 mM Tris, 250 mM NaCl, 0.08% digitonin, pH 7.8 (Binding Buffer A) Binding Buffer A + 3.5 mM desthiobiotin (Elution Buffer A) 20 mM Tris, 200 mM NaCl, 0.08% digitonin, pH 7.8 (Buffer B)	~3.0 Å Cryo-EM structure of trimeric human STEAP1 bound to three antigen- binding fragments of mAb 120.545 (PDB 6Y9B)	[5]
Recombinant Rabbit STEAP1	*Baculovirus*-Insect Cells	Ordinary Lysis Buffer (200 mM HEPES, 150 mM NaCl, 1 mM PMSF, 5 mM MgCl_2_, 5 mM Imidazole, 10 μM hemin chloride, 1.5% MNG-DDM, pH 7.5)	Affinity Chromatography (Talon Co^2+^) (*A*) Size Exclusion Chromatography (Superdex 200 10/300) (*B*)	20 mM HEPES, 150 mM NaCl, 20 mM Imidazole, 10 μM hemin chloride, 0.1% MNG-DDM, pH 7.5 (Buffer A) 20 mM HEPES, 150 mM NaCl, 0.01% MNG-DDM, pH 7.5 (Buffer B)	n.a.	[6]

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
