# Peer review of "Enhanced Stability of Detergent-Free Human Native STEAP1 Protein from Neoplastic Prostate Cancer Cells upon an Innovative Isolation Procedure"

_ijms, 2021, doi:10.3390/ijms221810012_

Round 1

Reviewer 1 Report

Good manuscript on “Enhanced Stability of Detergent-Free Human Native STEAP1 2 Protein from Neoplastic Prostate Cancer Cells Upon an Innovative Isolation Procedure” where authors showed a successfully method to isolate human STEAP1 from LNCaP cells. In addition authors should show the STEAP1 pathway regulation in prostate cancer. Also, The authors should check for the spelling in the text, and grammatical error, there are many mistakes.

Author Response

Dear Reviewer 1,

We appreciated the careful examination of the manuscript and greatly acknowledge the expertise comments provided by you. As you will see in the attached file of the manuscript, we have made the recommended changes.

Comment 1: In addition authors should show the STEAP1 pathway regulation in prostate cancer.”

Response 1: The authors acknowledge your suggestion about the STEAP1 pathway regulation in prostate cancer. Considering this observation, we added to the revised version of the manuscript the following sentence (Please see lines 50 to 56, page 2):

“Regarding the regulation of STEAP1 in PCa, it was demonstrated that 5α-dihydrotestosterone (DHT) down-regulates STEAP1 expression in LNCaP cells by androgen-independent receptor, suggesting that this down-regulation occurs in response to proliferation effects of DHT [9]. Also, it was shown that the knock-down of STEAP1 abrogated the proliferation and anti-apoptotic effects of DHT, indicating that blocking the STEAP1 protein can be advantageous in the treatment of hormone-dependent PCa [10].”

Comment 2: “Also, The authors should check for the spelling in the text, and grammatical error, there are many mistakes. Moderate English changes required”

Response 2: We kindly thank your attention calls, and we further revised the whole manuscript.

We hope that all the raised questions and all corrections performed over the manuscript have been properly clarified and that you consider this manuscript worthy of publication.

Let me close by thanking you in advance for the time spent in dealing with this manuscript.

Sincerely yours,

Luís António Paulino Passarinha

UCIBIO – Applied Molecular Biosciences Unit, Departamento de Química, Faculdade de Ciências e Tecnologia, Universidade NOVA de Lisboa, 2829-516 Caparica, Portugal

E-mail: [email protected] or [email protected]

Phone: +351 275 329 069, Fax: +351 275 319 883

Reviewer 2 Report

The paper titled "Enhanced Stability of Detergent-Free Human Native STEAP1 Protein from Neoplastic Prostate Cancer Cells Upon an Innovative Isolation Procedure” is interesting. The paper is well written and has the merit of publication. However, there are some issues that require attention:

  • Details about number of cells, replicate… should be provided
  • How do you compare with published data?
  • A resume integrative picture could be done to provide an illustration of data and transpose it.

Author Response

Dear Reviewer 2,

We appreciated the careful examination of the manuscript and greatly acknowledge the expertise comments provided by you. As you will see in the attached file of the manuscript, we have made the recommended changes.

Comment 1: “The paper is well written and has the merit of publication. However, there are some issues that require attention: Details about number of cells, replicate… should be provided”

Response 1: The authors acknowledge your suggestion for a more detailed explanation of the employed methodology. Then, we added to section “3.2. LNCaP Cell Culture” the following information (Please see lines 333 and 334, page 10):

“ (…) LNCaP cells have grown in 75 cm2 t-flasks (n = 6) until 80-90 % confluence for further harvest.”

Comment 2: “The paper is well written and has the merit of publication. However, there are some issues that require attention: How do you compare with published data? A resume integrative picture could be done to provide an illustration of data and transpose it.”

Response 2: We kindly thank your pertinent questions. Firstly, we added to the Introduction section, a recent study developed by our research team concerning the biosynthesis of recombinant STEAP1, aiming to update the published information with a very recent study (Please see lines 85 to 90, page 2):

“Recently, our research team proposed the application of a glycerol gradient fed-batch profile associated with a methanol constant feed, supplemented with 6 % (v/v) DMSO and 1 M Proline as an ideal fermentation strategy to improve the biosynthesis and stabilization of biological active recombinant human STEAP1 in mini-bioreactor Komagataella pastoris X-33 Mut+ cultures [22]. However, studies concerning the isolation and purification of this STEAP1 counterpart are still not available in the literature.”

Furthermore, we considered very useful for the reader to summarize all the existent information regarding this subject. So, we built a brief table with detailed information since the biosynthesis to the isolation and characterization of STEAP1 protein (Please see lines 126 to 135, pages 3 and 4):

“Considering the clinical relevance of STEAP1 and its potential application as a promising therapeutic agent against PCa, it is crucial to explore several expression systems, extraction strategies, and purification approaches to ultimately obtain a high-resolution 3D structure of the protein. Despite several attempts with the recombinant isoform (Table 1), experimental data about STEAP1 from its native cancer environment are still inexistent.

Table 1. Integrative overview of overall existent strategies from the biosynthesis to the isolation of STEAP1 counterparts towards their structural resolution and characterization.

Protein

Expression

System

Extraction

Isolation

Chromatographic Buffers

Structural Resolution

Ref.

Native

Human STEAP1

Neoplastic

Prostate

Cancer Cells

(LNCaP)

RIPA Buffer

(50 mM Tris Base,

150 mM NaCl,

1 mM EDTA,

0.5 % Sodium

Deoxycholate,

0.1 % SDS, 1 % NP-40,

pH 7.8)

Hydrophobic

Interaction

Chromatography

(Butyl-Sepharose)

coupled to

Co-Immunoprecipitation

1.375 M (NH4)2SO4, pH 7.8

(Binding)

10 mM Tris, pH 7.8

(Elution)

n.a.

This Work

Recombinant

Human STEAP1

Human

Embryonic

Kidney Cells (HEK)

Ordinary Lysis Buffer

(50 mM Tris,

250 mM NaCl,

0.7 % digitonin,

0.3 % n-Dodecyl-β-

D-Maltoside,

0.06% Cholesteryl

hemi-succinate, pH 7.8)

Affinity

Chromatography

(Streptactin) (A)

Size Exclusion

Chromatography

(Superdex 200 10/300) (B)

50 mM Tris,

250 mM NaCl, 0.08% digitonin, pH 7.8

(Binding Buffer A)

Binding Buffer A + 3.5 mM desthiobiotin

(Elution Buffer A)

20 mM Tris, 200 mM NaCl, 0.08 %

digitonin, pH 7.8 (Buffer B)

~3.0 Å

Cryo-EM structure of trimeric

human STEAP1 bound to three

antigen-

binding

fragments of mAb 120.545 (PDB 6Y9B)

[5]

Recombinant Rabbit STEAP1

Bacculovirus-Insect Cells

Ordinary Lysis Buffer

(200 mM HEPES, 150 mM NaCl, 1 mM PMSF,

5 mM MgCl2,

5 mM Imidazole,

10 μM hemin chloride,

1.5 % MNG-DDM,

pH 7.5)

Affinity

Chromatography

(Talon Co2+) (A)

Size Exclusion

Chromatography

(Superdex 200 10/300) (B)

20 mM HEPES,

150 mM NaCl,

20 mM Imidazole,

10 μM hemin chloride,

0.1 % MNG-DDM,

pH 7.5

(Buffer A)

20 mM HEPES,

150 mM NaCl,

 0.01 % MNG-DDM, pH 7.5

(Buffer B)

n.a.

[6]

We hope that all the raised questions and all corrections performed over the manuscript have been properly clarified and that may consider this manuscript worthy of publication.

Let me close by thanking you in advance for the time spent in dealing with this manuscript.

Sincerely yours,

Luís António Paulino Passarinha

UCIBIO – Applied Molecular Biosciences Unit, Departamento de Química, Faculdade de Ciências e Tecnologia, Universidade NOVA de Lisboa, 2829-516 Caparica, Portugal

E-mail: [email protected] or [email protected]

Phone: +351 275 329 069, Fax: +351 275 319 883 
